# 3DET-Mamba: State Space Model for End-to-End 3D Object Detection

**Mingsheng Li**[1,*], **Jiakang Yuan**[1,*], **Sijin Chen**[1,2], **Lin Zhang**[1],
**Anyu Zhu**[1], **Xin Chen**[2], **Tao Chen**[1,†]
[1] Fudan University    [2] Tecent PCG
* Equal contribution    † Corresponding author

## Abstract

Transformer-based architectures have been proven successful in detecting 3D objects from point clouds. However, the quadratic complexity of the attention mechanism struggles to encode rich information as point cloud resolution increases. Recently, state space models (SSM) such as Mamba have gained great attention due to their linear complexity and long sequence modeling ability for language understanding. To exploit the potential of Mamba on 3D scene-level perception, for the first time, we propose 3DET-Mamba, which is a novel SSM-based model designed for indoor 3D object detection. Specifically, we divide the point cloud into different patches and use a lightweight yet effective Inner Mamba to capture local geometric information. To observe the scene from a global perspective, we introduce a novel Dual Mamba module that models the point cloud in terms of spatial distribution and continuity. Additionally, we design a Query-aware Mamba module that decodes context features into object sets under the guidance of learnable queries. Extensive experiments demonstrate that 3DET-Mamba surpasses previous 3DETR on indoor 3D detection benchmarks such as ScanNet, improving AP@0.25/AP@0.50 from 65.0%/47.0% to 70.4%/54.4%, respectively.

## 1 Introduction

The aim of 3D object detection [24, 43, 26, 35] from point clouds is to locate and recognize objects present in 3D scenes. It is a challenging task since point clouds are often irregular, sparse, and unordered. To directly work with point clouds, VoteNet [31] utilizes PointNet++ [33] to extract features from irregular point clouds, which are then fed into a decoder to generate the 3D bounding boxes. Motivated by the success of Transformer [40] in computer vision [8, 34, 20, 3, 19], some works [24, 14, 26, 4] try to design Transformer-based 3D detectors. 3DETR [26] proposes an end-to-end transformer-based architecture to generate bounding boxes from raw point clouds. However, with limited computational resources, the quadratic complexity of the attention mechanism struggles to encode detailed representations, as it relies on increasing the point cloud resolution (*i.e.*, longer point cloud sequences).

Recently, state space models (SSMs) [10, 45, 16] have received significant attention due to their linear complexity and long-sequence modeling ability. As Mamba [10] demonstrates a strong ability to handle long sequences in natural language processing, it has rapidly been employed on different tasks (*e.g.*, image and 3D object classification [23, 60, 21, 22], video understanding [18, 1] and motion generation [55, 46]) and achieves great success. This motivates us to take advantage of Mamba in capturing long-range dependencies to extract more detailed representations in complex 3D scenes.

However, directly integrating Mamba [10] into the off-the-shelf detectors cannot achieve satisfactory results on 3D object detection tasks due to the following challenges. Firstly, SSMs like Mamba are causal models designed for handling 1-D sequence data, making it difficult to model unordered and

non-causal 3-D point clouds. Secondly, the original Mamba block focuses on modeling long-range global information but lacks the ability to extract local features, which are essential for point cloud learning [25, 29]. Besides, previous works primarily use Mamba as an encoder for single object analysis, it remains unexplored for more complex scene-level point clouds and detection tasks.

To tackle the above challenges and exploit the potential of Mamba [10] in 3D scene understanding, in this paper, we propose 3DET-Mamba, an end-to-end 3D detector that fully takes advantage of Mamba. To effectively extract scene representations from unordered point cloud sequences, we introduce a local-to-global scanning technique that can capture local geometry as well as global representation. Specifically, the local-to-global scanning technique utilizes the Inner Mamba block to capture finer details in each local patch and then uses Dual Mamba blocks to further extract scene features in a global view. Additionally, given that the naive Mamba block struggles to effectively model the relationship between object queries and scene features, we propose a query-aware Mamba block that decodes scene context information into object sets more effectively, guided by box queries. Extensive experiments on standard benchmarks show that our method can outperform 3DETR [26] on ScanNet and SUN RGB-D. Moreover, the performance can be further improved by increasing the input point cloud and learnable query sequences.

Our contribution can be summarized as follows:

- We introduce Mamba into 3D scene perception for the first time and construct an end-to-end detector named 3DET-Mamba which fully takes advantage of Mamba.

- We design a local-to-global scanning mechanism and develop the Inner Mamba and Dual Mamba, which account for both local detailed features and global spatial representations, respectively. Further, we propose a Query-aware Mamba to decode scene context features through learnable queries and generate bounding boxes for objects of interest.

- Extensive experiments demonstrate that 3DET-Mamba outperforms previous 3DETR on both the ScanNet and SUN RGB-D datasets, proving that Mamba can serve as a promising foundational component for 3D scene understanding in the future.

## 2 Related Work

In this section, we will briefly introduce existing works on 1) 3D object detection, and 2) state space models and Mamba, especially Mamba in vision tasks.

### 2.1 3D Object Detection

With the development of robotics, 3D object detection [31, 9, 2, 51, 52] from point clouds has attracted increasing attention. Existing works can be divided into grid-based and point-based methods. Grid-based (voxel-based) methods convert the irregular point clouds into 2D grids [17] or 3D voxels [42, 35, 36, 59, 50] to utilize the strong feature extraction ability of CNN. Among them, FCAF3D [35] proposes a fully convolution anchor-free framework. CAGroup3D [42] designs a two-stage pipeline by introducing class-aware proposal generation and RoI-conv pooling. However, these grid-based methods struggle to utilize the inherent sparsity of the data and incur significant computational expenses due to the 3D convolution operations [31]. Inspired by PointNet family [32, 33], Point-based methods [37, 26, 24, 31, 57, 48, 5] directly generate 3D proposals from raw points. As a pioneer, VoteNet [31] proposes a voting mechanism to generate accurate proposals by generating new points close to the objects' center. RBGNet [43] design a ray-based representation which largely improves the performance. Motivated by the success of Transformer on image detection, 3DETR [26] and Groupfree [24] introduce a transformer-based architecture and boost the detection accuracy. However, the quadratic complexity of the attention mechanism struggles to model long-range dependencies with limited computational resources. In this paper, we present a novel detection framework that can serve as a promising foundational module in 3D detection.

### 2.2 State Space Models and Mamba

**State Space Models and Mamba.** Inspired by the success in control theory, state space model (SSM) [13, 38, 28, 12] has been utilized to model long-range dependence. Structured State-Space Sequence (S4) [13] replaces CNN and Transformer with SSM in vision and language tasks by

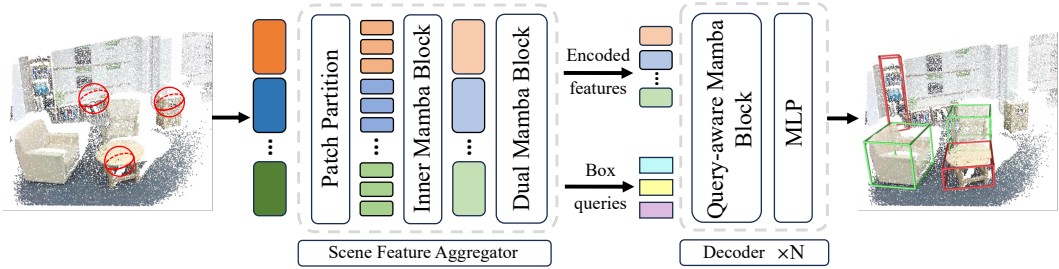

Figure 1: **The overview of the proposed 3DET-Mamba**. The point clouds are first patched and fed into the Inner Mamba block to learn fine-grained local features, which are then sent to the Dual Mamba block to extract global representations. These encoded scene information andd box queries go through the decoder, which includes Query-aware Mamba blocks and MLPs to generate the final bounding boxes. We employ the bipartite graph to match the predicted boxes with the ground truth and use a set loss for end-to-end optimization. Color is utilized only for visualization purposes.

combining linear SSM and HiPPO [11] framework. Smith *et al.* [38] propose S5 layer which designs a multi-input, multi-output SSM which outperforms S4 layer in both performance and efficiency. More recently, Mamba [10] with selective SSM is introduced which achieves higher performance than Transformer and leads to lots of further research on SSM [30, 44, 41]. For example, MoE-Mamba [30] and GraphMamba [41] combine Mamba with MoE and Graph data, respectively.

**Mamba in Vision.** Thanks to the breakthrough in natural language processing, Mamba has been rapidly transferred to various vision tasks [60, 23]. Vim[60] proposes a bi-directional SSM that can efficiently compress the vision representation and achieve satisfactory results on multiple vision tasks at low cost. VMamba [23] employs a cross-scan module to enable 1D selective scanning in 2D image space. Mamba-Unet [47] and MedMamba [53] introduce Mamba to medical image segmentation and classification, respectively. Video Mamba Suite [1] and VideoMamba [18] verify the effectiveness and efficiency of Mamba in various video tasks. QueryMamba [58] combines a query-based transformer decoder and the Mamba encoder to handle video action forecasting tasks. TM-Mamba [46] modifies the Mamba parameters as the function of the input and text query to ground the human motion. More recently, several works [54, 22, 21, 15] have explored the feasibility of Mamba on 3D tasks by introducing different point cloud ordering strategies. For example, PointMamba [21] utilizes Mamba to model the global information of 3D point clouds through a reordering mechanism and largely reduces the computation cost. Despite the great success achieved in object-level classification and part segmentation, Mamba in 3D scenes is under-explored. In this paper, we propose 3DET-Mamba which fills the gap of Mamba in 3D scene perception with designs like local-to-global scanning, dual Mamba, and query-aware Mamba.

## 3 Method

The overview framework of 3DET-Mamba is shown in Fig. 1. To better illustrate 3DET-Mamba, we first briefly review SSMs in Sec. 3.1. Then, we provide an overview of our 3DET-Mamba in Sec. 3.2. In Sec. 3.3 and Sec. 3.4, we detail the design of the encoder and decoder, respectively.

### 3.1 Preliminaries: State Space Models

State Space Models (SSMs) are derived from continuous systems and have been widely used in sequence modeling recently. Through a hidden state $h(t)$, SSMs can efficiently map 1D sequences input $x(t)$ to the output $y(t)$ using the following equations:

$$h'(t) = \mathbf{A}h(t) + \mathbf{B}x(t), \;\; y(t) = \mathbf{C}h(t), \tag{1}$$

where $\mathbf{A} \in \mathbb{R}$ denotes the evolution matrix, and $\mathbf{B} \in \mathbb{R}^{N \times 1}$, $\mathbf{C} \in \mathbb{R}^{1 \times N}$ are the projection matrices. To make SSMs can handle discrete signals such as language and point clouds, S4 [13] and Mamba [10] use a timescale parameter $\mathbf{\Delta}$ to transform the continuous parameters $\mathbf{A}$ and $\mathbf{B}$ into discrete ones $\overline{\mathbf{A}}$

and $\overline{\mathbf{B}}$. Specifically, Mamba uses zero-order hold (ZOH) method as follows:

$$\overline{\mathbf{A}} = \exp(\mathbf{\Delta A}), \ \ \overline{\mathbf{B}} = (\mathbf{\Delta A})^{-1}(\exp(\mathbf{\Delta A}) - \mathbf{I}) \cdot \mathbf{\Delta B}. \tag{2}$$

Then, the Eq. (1) can be reformulated to the discrete version as follows:

$$h_t = \overline{\mathbf{A}}h_{t-1} + \overline{\mathbf{B}}x_t, \ \ y_t = \mathbf{C}h_t. \tag{3}$$

Finally, a global convolution can be used to perform the model's calculation:

$$\overline{\mathbf{K}} = (\mathbf{C}\overline{\mathbf{B}}, \mathbf{C}\overline{\mathbf{A}}\overline{\mathbf{B}}, ..., \mathbf{C}\overline{\mathbf{A}}^{L-1}\overline{\mathbf{B}}), \ \ \mathbf{y} = \mathbf{x} * \overline{\mathbf{K}}, \tag{4}$$

where $L$ denotes the length of input sequence and $\overline{\mathbf{K}} \in \mathbb{R}^L$ presents a convolution kernel.

## 3.2 Overview

As shown in Fig. 1, our 3DET-Mamba takes a set of $N$ points $S \in \mathbb{R}^{N \times d}$ as input and generates a set of 3D (oriented) bounding boxes with semantic labels for all objects of interest. Specifically, 3DET-Mamba mainly consists of the Mamba-based scene feature aggregator and decoder designed for 3D object detection. The 3D feature aggregator combines Inner Mamba and Dual Mamba blocks to perform local-to-global scanning, extracting both the fine-grained local geometries and global contexts within the scene. On the decoding side, the decoder employs query-aware Mamba and Multi-Layer Perceptrons (MLPs) to decode the aggregated 3D features into discrete sets of objects. In the following sections, we will detail the designs of each component.

## 3.3 Scene Feature Aggregator

Recent advancements in point cloud processing have demonstrated the importance of capturing both local geometric features and global scene information [25]. However, previous methods primarily focus on global modeling using state-space models [21, 22, 54], resulting in a lack of fine-grained details. To tackle this challenge, we propose a novel local-to-global scanning technique that emphasizes scanning locally to uncover these finer details in each patch, followed by global scanning to capture the dependencies among local features.

In this section, we introduce a novel Mamba-based scene feature aggregator designed for local-to-global scanning, which is aimed at learning detailed representations of complex 3D scenes, as illustrated in Fig. 2. Specifically, our scene feature aggregator includes the Inner Mamba block and Dual Mamba block, which will introduced in Sec. 3.3.1 and Sec. 3.3.2 respectively.

### 3.3.1 Inner Mamba

Given an unordered $N$ point cloud sequence $\{x_1, x_2, \ldots, x_n\}$, where $x_i \in \mathbb{R}^d$ and $d$ is the dimension of features, farthest point sampling (FPS) is first used to select $K$ key points from the input point cloud. Then k-nearest neighbors (KNN) is utilized to identify $N_0$ nearest neighbors for each key point, subsequently forming $K$ patches $P \in \mathbb{R}^{K \times N_0 \times d}$.

Previous works [33, 21, 22] employ MLPs to learn features from each patch, they struggle with the effective aggregation of local features. To address this, we treat the aggregation of local features as a sequence-to-sequence generation process and use a causal Mamba model to extract local features within a patch. Specifically, points within each patch are first normalized and then sorted based on their distance from the key point. These sequential points are fed into a lightweight Mamba block, of which the dimensions are reduced compared to the original Mamba block, to generate a new feature sequence. Finally, max-pooling is used to obtain the embedding for each patch, denoted as $F^s \in \mathbb{R}^{K \times C}$, where $K$ denotes the number of patches and $C$ is the embedding dimension.

### 3.3.2 Dual Mamba

To encode point cloud data, previous works such as PointMamba [21] propose learning 3D representations by utilizing the original Mamba, which is initially designed for 1-D ordered sequences. Since understanding 3D data requires capturing global information, Point Cloud Mamba [54] suggests reversing the order of tokens and employing both forward and backward State Space Models (SSMs) to better capture the global context. However, due to the unordered and irregular nature of point

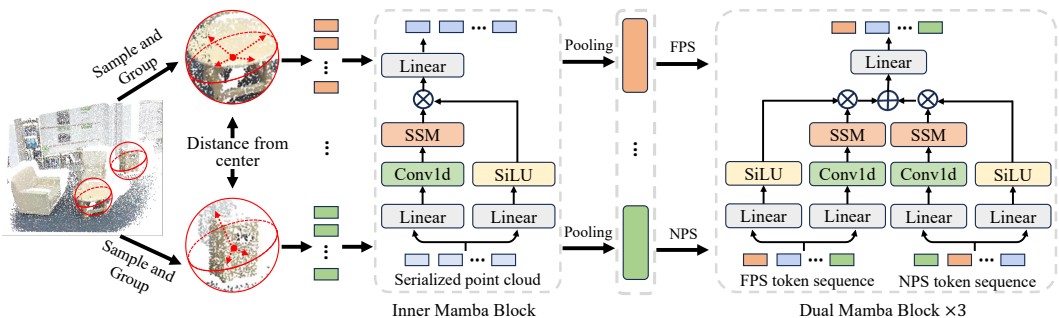

Figure 2: **Architecture of our scene feature aggregator that employs a novel local-to-global scanning mechanism.** The raw point clouds are first sampled and patched using FPS and KNN. Within each patch, points are ranked by their distance from the patch center. The Inner Mamba block then scans these ranked points to extract local geometric features. Subsequently, patches are treated as tokens and serialized in two manners before being fed into the Dual Mamba block. This step scans all tokens, extracting comprehensive scene contexts.

---

**Algorithm 1** Dual Mamba Block

---

**Input:** token sequence $\mathbf{T}_{l-1} : (\mathtt{B,K,C})$
**Output:** token sequence $\mathbf{T}_l : (\mathtt{B,K,C})$

1: /* process with different point orders */
2: $\mathbf{T}_{l-1}^F : (\mathtt{B,K,C}) \leftarrow \mathbf{T}_{l-1}$
3: $\mathbf{T}_{l-1}^N : (\mathtt{B,K,C}) \leftarrow \mathbf{NPS}(\mathbf{T}_{l-1})$
4: **for** $o$ in $\{F, N\}$ **do**
5: $\quad \mathbf{T}_{l-1}^{o}{}' \leftarrow \mathbf{Norm}(\mathbf{T}_{l-1}^o)$
6: $\quad \mathbf{z}_o : (\mathtt{B,K,C'}) \leftarrow \mathbf{Linear}_{\mathbf{z}_o}(\mathbf{T}_{l-1}^{o}{}')$
7: $\quad \mathbf{x}_o : (\mathtt{B,K,C'}) \leftarrow \mathbf{Linear}_{\mathbf{x}_o}(\mathbf{T}_{l-1}^{o}{}')$
8: $\quad \mathbf{x}_o' : (\mathtt{B,K,C'}) \leftarrow \mathbf{SiLU}(\mathbf{Conv1d}_o(\mathbf{x}_o))$
9: $\quad \mathbf{B}_o : (\mathtt{B,K,D}) \leftarrow \mathbf{Linear}_o^{\mathbf{B}}(\mathbf{x}_o')$
10: $\quad \mathbf{C}_o : (\mathtt{B,K,D}) \leftarrow \mathbf{Linear}_o^{\mathbf{C}}(\mathbf{x}_o')$
11: $\quad$ /* softplus ensures positive $\boldsymbol{\Delta}_o$ */
12: $\quad \boldsymbol{\Delta}_o : (\mathtt{B,K,C'}) \leftarrow \log(1 + \exp(\mathbf{Linear}_o^{\boldsymbol{\Delta}}(\mathbf{x}_o') + \mathbf{Parameter}_o^{\boldsymbol{\Delta}}))$
13: $\quad$ /* shape of $\mathbf{Parameter}_o^{\mathbf{A}}$ is $(\mathtt{C',D})$ */
14: $\quad \overline{\mathbf{A}}_o : (\mathtt{B,K,C',D}), \overline{\mathbf{B}}_o : (\mathtt{B,K,C',D}) \leftarrow \mathbf{Disc}(\boldsymbol{\Delta}_o, \mathbf{Parameter}_o^{\mathbf{A}}, \mathbf{B}_o)$
15: $\quad \mathbf{y}_o : (\mathtt{B,K,C'}) \leftarrow \mathbf{SSM}(\overline{\mathbf{A}}_o, \overline{\mathbf{B}}_o, \mathbf{C}_o)(\mathbf{x}_o')$
16: $\quad$ /* get gated $\mathbf{y}_o$ */
17: $\quad \mathbf{y}_o' : (\mathtt{B,K,C'}) \leftarrow \mathbf{y}_o \odot \mathbf{SiLU}(\mathbf{z}_o)$
18: **end for**
19: $\mathbf{y}_N' : (\mathtt{B,K,C'}) \leftarrow \mathbf{FPS}(\mathbf{y}_N')$
20: $\mathbf{T}_l : (\mathtt{B,K,C}) \leftarrow \mathbf{Linear}^{\mathbf{T}}(\mathbf{y}_F' + \mathbf{y}_N') + \mathbf{T}_{l-1}$
21: Return: $\mathbf{T}_l$

---

clouds, simply reversing the order of tokens cannot ensure the causal dependency of the point cloud sequence and may lead to unreliable results.

To mitigate such a problem, we introduce a novel Dual Mamba block (as shown in Fig. 2) to model long-range dependencies from the global view. Specifically, we treat $F^s \in \mathbb{R}^{K \times C}$ as tokens and sort them based on their coordinates into two categories: farthest and nearest neighbor orders. The former strategy enhances the model's perception of spatial distribution by maximizing the distances between adjacent points in the sequence, whereas the latter ensures that adjacent points remain spatial neighbors, thus maintaining local consistency. The Dual Mamba block incorporates two branches to handle FPS and NPS token sequences, which first undergo normalization and independent linear projection. For each sequence, an initial 1D convolution transforms it into $\mathbf{x}_o'$, which is then projected

into $\mathbf{A}_o$, $\mathbf{B}_o$, and $\mathbf{\Delta}_o$. Subsequently, $\mathbf{A}_o$ and $\mathbf{B}_o$ are discretized using $\mathbf{\Delta}_o$. Finally, the corresponding tokens from the two branches are added and passed through a linear layer to generate the scene representations. The specific details are shown in Algorithm 1.

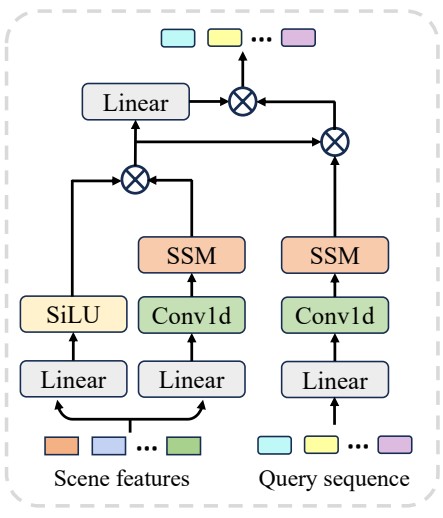

Figure 3: **Query-aware Mamba block.**

**Algorithm 2** Query-aware Mamba Block

**Input:** 3D scene features $\mathbf{F^s_{t-1}}$: $(\mathbf{B}, \mathbf{K}, \mathbf{C})$,
Box query sequence $\mathbf{F^q_{t-1}}$: $(\mathbf{B}, \mathbf{M}, \mathbf{C})$
**Output:** Query sequence $\mathbf{F^q_t}$: $(\mathbf{B}, \mathbf{M}, \mathbf{C})$
1: **for** o **in** $\{\mathbf{s}, \mathbf{q}\}$ **do**
2:     $F^o_{t-1}{}' \leftarrow \mathbf{Norm}(F^o_{t-1})$
3:     $\mathbf{x_o}$: $(\mathbf{B}, -, \mathbf{C'}) \leftarrow \mathbf{Linear_{x_o}}(\mathbf{F^o_{t-1}}{}')$
4:     $\mathbf{x'_o}$: $(\mathbf{B}, -, \mathbf{C'}) \leftarrow \mathbf{SiLu}(\mathbf{Conv_o}(\mathbf{x_o}))$
5:     /* **Disc** and **SSM** */
6:     $\overline{\mathbf{A}}$: $(\mathbf{B}, -, \mathbf{C'}, \mathbf{D})$, $\overline{\mathbf{B}}$: $(\mathbf{B}, -, \mathbf{C'}, \mathbf{D})$,
      $\mathbf{C}$ : $(\mathbf{B}, -, \mathbf{C'}, \mathbf{D}) \leftarrow \mathbf{Disc}(\mathbf{x'_o})$
7:     $\mathbf{y_o}$: $(\mathbf{B}, -, \mathbf{C'}) \leftarrow \mathbf{SSM}(\overline{\mathbf{A}}, \overline{\mathbf{B}}, \mathbf{C})(\mathbf{x'_o})$
8:     $\mathbf{z}$ : $(\mathbf{B}, -, \mathbf{C'}) \leftarrow \mathbf{Linear_z}(\mathbf{F^s_{t-1}})$
9:     $\mathbf{y'_o}$: $(\mathbf{B}, -, \mathbf{C'}) \leftarrow \mathbf{y_o} \odot \mathbf{SiLu}(\mathbf{z})$
10: **end for**
11: $\mathbf{y'_q}$: $(\mathbf{B}, \mathbf{M}, \mathbf{K}) \leftarrow \mathbf{y_q} \odot \mathbf{y'_s}$
12: $\mathbf{F^q_t}$: $(\mathbf{B}, \mathbf{M}, \mathbf{C}) \leftarrow \mathbf{Linear_F}(\mathbf{y'_s}) \odot \mathbf{y'_q}$
13: $\mathbf{F^q_t}$: $(\mathbf{B}, \mathbf{M}, \mathbf{C}) \leftarrow \mathbf{F^q_t} + \mathbf{Norm}(\mathbf{F^q_{t-1}})$
14: **return** $\mathbf{F^q_t}$;

## 3.4 Decoder

DETR-based models leverage a set of object queries to extract features for object classification and localization. However, directly using scene context as a prefix and concatenating it with queries in the Mamba model leads to suboptimal performance, as it struggles to capture discriminative features for independent queries. To address this, we introduce the Query-aware Mamba block for the decoder, as shown in Fig. 3, which effectively models the relationship between learnable queries and scene features to generate bounding boxes.

We first generate object queries by selecting a defined number of $M$ points from the set of $K$ key points using FPS, ensuring these query points cover the entire scene. For each of these points, we follow 3DETR [26] to transform their spatial coordinates into positional embeddings using the Fourier Transform. These embeddings are subsequently processed through an MLP to produce the initialized query embeddings.

In detail, the decoder is composed of $D$ same blocks and each block consists of a query-aware Mamba block and multiple MLP layers. The Query-aware Mamba block takes box queries and scene context as input, extracting tasked-related features from the scene context guided by the learnable queries. Specifically, each query sequence $\mathbf{F^q}$ is fed into a standard Mamba block to model the dependencies between queries. This process can be formulated as follows:

$$\begin{aligned}\mathbf{F^q_o} &= \mathbf{Linear}(\mathbf{Norm}(\mathbf{F^q})) \\ \mathbf{F^q} &= \mathbf{SiLU}(\mathbf{F^q_o}) \times \mathbf{SSM}(\mathbf{Conv}(\mathbf{F^q_o})).\end{aligned} \tag{5}$$

Meanwhile, scene features undergo the same process as the query sequence. Then, by multiplying the scene features with query embeddings, scene contexts are integrated into the query embeddings, and the updated queries are then passed through multiple MLP layers. After $D$ blocks decoding process, the bounding boxes and semantic categories can be generated using MLP-based heads. Please refer to the Algorithm 2 for more details.

## 3.5 Training Objectives

We adopt the same training objectives as [26] to train 3DET-Mamba. Specifically, we use the bipartite graph to match the set of predicted 3D bounding boxes $\{\hat{b}\}$ with the ground truth boxes $\{b\}$, denoted

as $L_{giou}(\hat{b}, b)$. Then we calculate the discrepancies between $\{\hat{b}\}$ and $\{b\}$ using $\ell_1$ loss for centers and dimensions, and Huber loss for angular residuals:

$$L_{geo} = \lambda_c\|\hat{c} - c\|_1 + \lambda_d\|\hat{d} - d\|_1 + \lambda_a\|\hat{a}_r - a_r\|_{huber}, \tag{6}$$

here, $\lambda_c$, $\lambda_d$, $\lambda_a$ are set as 5, 1, and 0.5. We also employ cross-entropy losses to assess angular and semantic classifications:

$$L_{sem} = -\lambda aca_c \log \hat{a}_c - \lambda_s s \log \hat{s}, \tag{7}$$

here, $\lambda_{ac}$ and $\lambda_s$ are set as 0.1 and 1. Finally, the total loss is formulated as:

$$L_{3DET\text{-}Mamba} = L_{giou} + L_{geom} + L_{sem}. \tag{8}$$

## 4 Experiments

In this section, we first describe our experiment setups, including our benchmark datasets, metrics, and implementation details in Sec. 4.1. Then we present our main results in Sec. 4.2 and take out ablation studies to analyze the effectiveness of the proposed component in Sec. 4.3. Finally, we showcase some visualization results in Section Sec. 4.4.

### 4.1 Datasets, Metrics, and Implementation Details

**Datasets.** Following previous works on 3D indoor object detection, we evaluate our models on two challenging benchmarks: SUN RGB-D [39] and ScanNet [7]. The SUN RGB-D [39] dataset consists of 10,335 single-view RGB-D scans, with 5,285 used for training and 5,050 for validation. Each sample is annotated with rotated 3D bounding boxes. Following [31, 26], we convert the RGB-D images into point clouds using camera parameters and evaluate models on the 10 most common object categories. ScanNet [7] comprises 1,201 training samples and 312 validation samples, with each sample annotated with axis-aligned bounding box labels for 18 object categories.

**Metrics.** Following [24, 26], we use standard evaluation protocols [31, 35] and report the detection performance on the validation set using mean Average Precision (mAP) at two different IoU thresholds (*i.e.*m 0.25 and 0.5), denoted as mAP@0.25 and mAP@0.5, respectively.

**Implementation Details.** The input to our model is a point cloud $P \in \mathbb{R}^{N \times 3}$ representing a 3D scene, with $N$ set as 20,000 for SUN RGB-D [39] and 40,000 for ScanNet [7]. We employ a single-layer inner mamba block that generates 2048 patches, each with 256-dimensional features. The dual mamba encoder has 3 layers and outputs scene features with a hidden dimension of 256. The decoder has 8 layers and is closely followed by MLPs as the bounding box prediction head. During training, we employ standard data augmentation methods, including random cropping, sampling, and flipping. We use the AdamW optimizer with a base learning rate of $7 \times 10^{-4}$, decayed to $10^{-6}$ using a cosine schedule, and a weight decay of 0.1. Gradient clipping with an $\ell_2$ norm of 0.1 is applied to stabilize training. The whole model is implemented in PyTorch, and all experiments are conducted on 8 NVIDIA 3090 GPUs (24 GB) with a total batch size of 64.

### 4.2 Comparisons on 3D Object Detection

In this section, we compare our 3DET-Mamba with previous 3D detectors. As shown in Tab. 1, our 3DET-Mamba can outperform previous 3DETR [26] (Tramsformer-based detectors) on both SUN RGB-D and ScanNet datasets. For example, with 256 queries and 2048 points, our method achieves 66.9% mAP@0.25 and 48.7% mAP@0.5 on ScanNet, surpassing 3DETR-m, which obtains 65.0% mAP@0.25 and 47.0% mAP@0.5 respectively. Besides, since Mamba can effectively handle long sequences, we further conduct experiments using point clouds with higher resolution and more box queries (*i.e.*, 4096 point clouds and 512 box queries), we can observe that the performance can be further improved with longer point and query sequences (*i.e.*, +5.7 mAP@0.5).

### 4.3 Ablation Studies

**Analysis of the encoder.** To verify the effectiveness of the designed Mamba-based encoder, we first conduct ablation studies on Inner Mamba and Dual Mamba. Specifically, we replace Inner Mamba and Dual Mamba with Pointnet++ and Transformer. As reported in Tab. 2, transforming Inner

Table 1: 3D detection results on ScanNet V2 [7] and SUN RGB-D [39]. We compare 3DET-Mamba against open-source methods that directly process point clouds, using PointNet++[33] (PN) and Transformer[40] (Tran.) as backbones. 3DET-Mamba employs the same number of key points and queries as 3DETR [26]. 3DET-Mamba† doubles the number of key points and queries to model longer sequences of point clouds. Compared to 3DETR [26], 3DET-Mamba achieves superior performance.

| Methods | Backbone | ScanNet | | SUN RGB-D | |
|---|---|---|---|---|---|
| | | mAP@0.25 | mAP@0.5 | mAP@0.25 | mAP@0.5 |
| VoteNet [31] | PN | 58.6 | 33.5 | 57.7 | - |
| MLCVNet [49] | | 64.5 | 41.4 | 59.8 | - |
| H3DNet [56] | | 67.2 | 48.1 | 60.1 | 39.0 |
| BRNet [6] | | 66.1 | 50.9 | 61.1 | 43.7 |
| GroupFree3D [24] | | 67.3 | 48.9 | 63.0 | 45.2 |
| GroupFree3D* [24] | | 69.1 | 52.8 | - | - |
| 3DETR [27] | Tran. | 62.7 | 37.5 | 58.0 | 30.3 |
| 3DETR-m [27] | | 65.0 | 47.0 | 59.1 | 32.7 |
| 3DET-Mamba | Mamba | 66.9 | 48.7 | 59.3 | 33.4 |
| 3DET-Mamba† | | **70.4** | **54.4** | **61.3** | **42.2** |

Table 2: Effect of our designed point cloud encoder which contains Inner and Dual Mamba blocks. We compare the performance of encoder combinations using Pointnet++ [33] and Transformer [26] with Inner and Dual Mamba blocks, with results showing that the proposed Inner Mamba & Dual Mamba achieves the best performance.

| Encoder | mAP@0.25 | mAP@0.5 |
|---|---|---|
| Pointnet++ & Transformer | 63.1 | 44.1 |
| Inner Mamba & Transformer | 64.9 | 45.4 |
| Pointnet++ & Dual Mamba | 65.6 | 48.4 |
| Inner Mamba & Dual Mamba | **66.9** | **48.7** |

Mamba to PointNet++ results in a performance drop of 1.8% mAP@0.25 and 1.3% mAP@0.5 when using Transformer and Dual Mamba, respectively. This is because our Inner Mamba can effectively aggregate and propagate local features. Besides, our Dual Mamba block can bring +4.3% mAP@0.5 and +3.3% mAP@0.5 improvement compared to using Transformer as the spatial encoder since our Mamba layers can learn both spatial representation and local consistency at the same time. By combining the Inner and Dual Mamba, the best results can be achieved which further verifies our designs.

**Effect of Dual Mamba block.** To further show the advantage of our designed Dual Mamba block, we replace the Dual Mamba block with the original Mamba [10] block and Bi-Mamba block [60, 22, 54], respectively. The original Mamba block only contains a forward SSM and the Bi-Mamba block is composed of a forward and a backward SSM. As shown in Tab. 3, our proposed dual Mamba block can outperform both the original Mamba block and Bi-Mamba block by +1.0% and +1.2% mAP@0.5. This is because our dual Mamba block can provide point cloud sequence with short-range and long-range dependencies and further make better use of the powerful causal modeling ability of Mamba.

**Effect of Query-aware Mamba.** We conduct ablation studies on the decoder to demonstrate the effectiveness of the designed Query-aware Mamba block. We compare our module against the following baselines: (a) a transformer-based decoder as proposed in 3DETR [26]; (b) a naive approach that directly concatenates 3D scene information with queries, feeding them into the Mamba block and leveraging Mamba's state transition matrix for further modeling; and (c) our proposed Query-aware Mamba block. Additionally, all experiments are conducted with the same encoder and training strategy to ensure a fair comparison. As demonstrated in Tab. 4, the decoder based on the Query-aware Mamba block achieves superior results compared to both baselines. This improvement can be attributed to two main factors: (1) Directly concatenating queries and 3D scene representations do

Table 3: Effect of Dual Mamba block. We compare it with the original Mamba block [10] and the Bi-directional Mamba block [60, 22].

| Mamba Block | mAP@0.25 | mAP@0.5 |
|---|---|---|
| Ori. Mamba | 65.4 | 47.7 |
| Bi. Mamba | 66.4 | 47.5 |
| Dual Mamba | **66.9** | **48.7** |

Table 4: Effect of Query-aware Mamba. We compare it with the 3DETR [26] decoder and an implementation based on the original Mamba.

| Decoder | mAP@0.25 | mAP@0.5 |
|---|---|---|
| Transformer | 64.3 | 42.6 |
| Ori. Mamba | 56.6 | 28.0 |
| Quety-aware Mamba | **66.9** | **48.7** |

not effectively model the relationship between them, making it difficult to capture the most relevant 3D features for each query. (2) Our Query-aware Mamba decoder more effectively aggregates dependencies between queries and scene features, enhancing the extraction of task-relevant features from the context.

### 4.4 Qualitative Results.

To demonstrate the effectiveness of 3DET-Mamba more intuitively, we show some visualization results. As shown in Fig. 4, we can observe that our 3DET-Mamba can accurately detect objects.

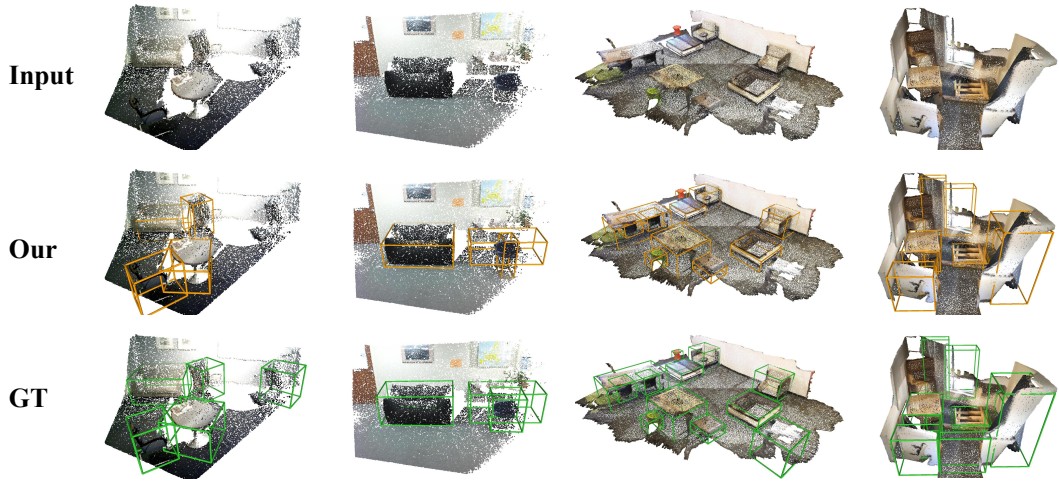

Figure 4: **Visualization of detection results.** 3DET-Mamba is able to generate tight bounding boxes for objects of interest in these complex and diverse scenes.

## 5 Conclusion

In this paper, for the first time, we exploit the potential of Mamba in 3D object detection tasks. Specifically, we introduce 3DET-Mamba, an end-to-end detector based on encoder-decoder architecture. We first propose an SSM-based encoder that uses an Inner Mamba block to capture local geometric information and uses Dual Mamba blocks to further aggregate features in a global view. Besides, a Query-aware Mamba module is designed to effectively decode scene representations into object sets with the guide of learnable box queries. Extensive experiments on standard benchmarks like ScanNet and SUN RGB-D demonstrate the effectiveness of 3DET-Mamba, proving Mamba as a promising building block for future 3D scene understanding. In addition, with the increased length of input point sequence and query sequence, the performance will be further boosted since Mamba is an expert in long sequence modeling.

## 6 Limitations and Broader Impacts

While 3DET-Mamba has demonstrated effectiveness in modeling point cloud sequences, we have yet to explore its potential for handling other 3D data types, such as meshes. Additionally, given Mamba's

strength in modeling long-sequence data, a promising future direction is to develop Mamba-based 3D foundation models capable of addressing a broader range of scene-level tasks, including 3D dense caption, visual grounding, and 3D QA. We leave these explorations for future work.

## Acknowledgement

This work is supported by National Natural Science Foundation of China (No. 62071127, and 62101137), National Key Research and Development Program of China (No. 2022ZD0160101), Shanghai Natural Science Foundation (No. 23ZR1402900), Shanghai Municipal Science and Technology Major Project (No.2021SHZDZX0103). The computations in this research were performed using the CFFF platform of Fudan University.

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
