# OpenReview forum: "3DET-Mamba: Causal Sequence Modelling for End-to-End 3D Object Detection"
_NeurIPS.cc/2024/Conference — NeurIPS 2024 poster_

### Official Review · Reviewer_eAYh · 2024-07-11

**Soundness:** 3
**Presentation:** 3
**Contribution:** 3
**Rating:** 6
**Confidence:** 4

**Summary:**

This paper proposes 3DET-Mamba, the first attempt to exploit the State Space Model for end-to-end 3D object detection. It introduces a local-to-global scanning technique, including an Inner Mamba block to capture local geometry and Dual Mamba blocks to extract scene features in a global view. Furthermore, it proposes a query-aware Mamba block to effectively decode scene context information into object sets with the guidance of box queries. Experiments show promising results compared to 3DETR and validate the effectiveness of the proposed modules with detailed ablation studies.

**Strengths:**

- The basic idea is easy to follow, and the key challenges are clearly presented.
- The methodology and implementation details are also clear, with good illustrations and mathematical / algorithm presentations.
- The proposed framework is the first to successfully adopt Mamba in end-to-end 3D object detection.
- The experimental results and ablation studies are solid and can support the method convincingly.

**Weaknesses:**

- One of the motivations is the computational costs of transformers compared to Mamba, but there are no analyses regarding memory costs, latency, etc. I am curious about more advantages of the Mamba framework in this task, such as computational efficiency and scaling up performance, except for the current traditional benchmarks.
- Although the paper proposes several methods to deal with the unordered point clouds in the ordered manner that is inherently designed in the State Space Model, even though the performance also seems good, the mechanism is still not most suitable for this task or handling point clouds. I would expect a more matching mechanism to deal with point clouds with the fundamental idea of the State Space Model instead of converting the problem with a non-perfect workaround.
- There are several typos and I just provide a few examples in Questions. The paper needs more check on such details.

**Questions:**

Sec. 4.2 title: Obejct -> Object
line 241: Transforme -> Transformer

**Limitations:**

None.

---

> ### Author Rebuttal · Authors · 2024-08-07
>
> **Q1**: Computational costs and scaling up performance.
>
> **A1**:
> - To further verify the effectiveness of 3DET-Mamba, we report latency and memory costs in the following table. It can be seen that our model achieves better results with less computational cost and lower latency compared to 3DETR. The results further verify the effectiveness of 3DET-Mamba.
> - Besides, we conduct more experiments in which we scale up the model size (more mamba block). As shown in Figure 2 and Figure 3 in the PDF file, both scaling the encoder and decoder can further improve the performance on ScanNet which further shows the scaling ability of 3DET-Mamba.
>
> **Performance Comparison of Computational Efficiency**
> | Method          | FLOPs (↓) | Latency (↓) |
> |-----------------|-----------|-------------|
> | 3DETR           | 14.3      |        0.22     |
> | 3DET-Mamba (our)| 9.8       |     0.13        |
>
> **Q2**: Discussion on the proposed method.
>
> **A2**:
> - In this paper, we first explore the potential of Mamba in 3D object detection tasks which have not been studied. However, directly using mamba to deal with 3D object detection tasks results in poor results. This is mainly caused by 1) It is difficult for Mamba to model unordered and non-causal 3-D point clouds, 2) the original Mamba block lacks the ability to extract local features, and 3) previous works only explore Mamba as an encoder for classification tasks.
> - To handle the above challenges, we propose a local-to-global scanning technique which is composed of Inner Mamba blocks and dual Mamba blocks and can aggregate local and global information. Besides, in dual Mamba blocks, we use furthest point sampling and nearest point sampling to construct the ordered point cloud sequence to handle the non-casual problem. Finally, the query-aware Mamba block is designed to decode scene context information into object sets with the guidance of box queries.
> - We hope our work can Inspire further exploration of mamba in 3D detection tasks and construct Mamba-base 3D foundation models in the future.
>
> **Q3**: Some typos.
>
> **A3**: Thanks for carefully reviewing our article. We will modify all these typos in the next version.

---

> > ### Comment · Reviewer_eAYh · 2024-08-09
> > **Final Decision**
> >
> > Thanks for the author's response. I think the experiments regarding the computational efficiency are important supplements to support the method's value, but there are still concerns regarding whether the Mamba structure is suitable for solving the problem with unordered point clouds input, both from theoretical analysis and intuition. It also lacks more comparison with state-of-the-art voxel-based methods like FCAF3D in the experiments. Therefore, I agree that this paper, as the first attempt, still has a long way to go along this technical pathway, but I may give more credit to the first try with basically reasonable experimental results as the support, and I think it can bring some new insights to the community. Hence, I would keep my original rating.

---

> ### Author Response · Authors · 2024-08-13
>
> Dear reviewer,
>
> Thank you for taking the time to review our work and for your feedback! We will add experimental results of computational efficiency in the revision of our article.
>
> Sincerely,
> Authors of paper112

---

### Official Review · Reviewer_2ZE6 · 2024-07-12

**Soundness:** 3
**Presentation:** 2
**Contribution:** 3
**Rating:** 5
**Confidence:** 5

**Summary:**

The paper proposes an end-to-end 3D detector named 3DET-Mamba that fully takes advantage of Mamba. 3DET-Mamba can model long-range global information (Dual Mamba) while exploiting local information (Inner Mamba). Experiments conducted on the ScanNet and SUN RGB-D datasets validate the effectiveness of the proposed method.

**Strengths:**

- The proposed method is simple and effective: 1) The use of distances from the center to rank points and query features is intriguing and straightforward to implement; 2) Employing FPS and NPS simultaneously to generate token sequences appears to be a novel approach, effectively modeling long-range dependencies while maintaining local consistency.
- The writing is clear and concise, making it easy to understand.
- The experimental results are impressive.

**Weaknesses:**

- The structure depicted in Figure 3 is inconsistent with the description provided in Algorithm 2. Specifically, the branch for the Query sequence in Figure 3 lacks the inclusion of Linear-SiLU, which is mentioned in Algorithm 2.
- In Table 4, 1) the first column should correctly be labeled as "mAP@0.25," and the second column should be labeled as "mAP@0.5." 2) when using the Transformer as a decoder, the value of mAP@0.5, which is 42.6%, appears abnormal compared to other results. Specifically, in Table 1, the model DETR-m achieved 65.0% mAP@0.25 and 47.0% mAP@0.5. However, in Table 4, the first model listed achieves 64.3% mAP@0.25 but only 42.6% mAP@0.5. Are there any mistakes?
- Line 241 `PointNet++ and Transforme` -> 'PointNet++ and Transformer'.
- Line 180-181, `We randomly choose an initial point and sort the remaining points based on their distance to this point`, are the experimental results stable? It seems unbelievable.

**Questions:**

See the Weaknesses.

---

> ### Author Rebuttal · Authors · 2024-08-07
>
> **Q1**: Difference between Algorithm 2 and Figure 3.
>
> **A1**: Thanks for reviewing our article carefully. The algorithm 2 shows the detailed operation steps of our model. The figure 3 is a schematic figure and may omit some details. According to the suggestion of the reviewer, we will modify Figure 3 in the revised version.
>
> **Q2**: Question about Table 4.
>
> **A2**:
> - Thanks for pointing out our mistakes. We will modify the headers of Table 4 in the revised version.
> - We want to clarify that in Table 1, we report the results of 3DETR in the original paper which is trained for 1080 epochs. However, in Table 4, for a fair comparison, we only change the decoder to 3DETR decoder and use our mamba-based encoder. We train for 540 epochs which is consistent with 3DET-Mamaba. We will clarify it in the next version.
>
> **Q3**: Some typos.
>
> **A3**: Thanks for your comments. We will revise all the typos in the next version.
>
> **Q4**: Question about randomly choosing the initial point.
>
> **A4**: The aim of this paper is to explore the potential of Mamba in 3D object detection. We follow 3DETR[A] to random sample an initial point and then sample a set of points using FPS which are evenly distributed in 3D space. Besides, based on our experiments, the experimental results are stable and we repeat the experiments. The results are shown in the following table. Since the difference between the random seed and the GPU device, there is a slight deviation from the submitted article. However, it can be observed that the experimental results are still stable.
>
> **Repeated experimental results of 3DET-Mamba**
> |Method|mAP25|mAP50|
> |-|-|-|
> |Result in the main paper|66.9|48.7|
> |Replicate|66.3|48.2|
>
> [A] Misra I, Girdhar R, Joulin A. An end-to-end transformer model for 3d object detection[C]//Proceedings of the IEEE/CVF international conference on computer vision. 2021: 2906-2917.

---

### Official Review · Reviewer_m4H4 · 2024-07-13

**Soundness:** 2
**Presentation:** 3
**Contribution:** 2
**Rating:** 5
**Confidence:** 4

**Summary:**

This paper proposes leveraging Mamba blocks for 3D point cloud modeling in the form of 3DET-Mamba, an application of SSM for 3D object detection. Similar to the prior 3DETR model, this approach partitions the point cloud into "patch" using Mamba blocks to capture local information, complemented by global modeling through dual Mamba blocks (organized by farthest and nearest point order). Additionally, query-aware Mamba blocks, akin to transformer blocks, are designed to decode objects in a DETR-like manner. The novelty of the approach lies in its ability to enhance 3DETR by integrating SSMs innovatively for 3D indoor scene understanding, utilizing ScanNet and SUN RGB-D datasets.

**Strengths:**

1. The proposed architecture adopts a Mamba-style approach and introduces 3D-specific customizations, such as partitioning the point clouds using an inner Mamba network, employing FPS/NPS ordering of points, and implementing a query-aware Mamba block.
2. The experiments detailed in Tables 2, 3, and 4 conduct ablation studies that demonstrate the effectiveness of the Mamba block compared to baseline designs using transformers, although these comparisons do not include assessments of speed.

**Weaknesses:**

1. Mamba is praised for handling linear complexity and long-range sequences better than transformers, which have higher computational demands. However, the paper lacks theoretical analysis or experimental evidence to compare time and space complexities between Mamba and traditional architectures. The effects of changing point resolutions or patch sizes on performance are also not explored.

2. The clarity of some ablation studies is lacking. For example, the role of ranked queries, a key contribution, is not evaluated. The relevance of NPS and FPS ordering in modeling patch sequences remains ambiguous. Additionally, the rationale for query-aware Mamba blocks is unconvincing; they seem to be simply added to cross-attention blocks with extra links. Table 4 does not clearly show that improvements are due to the Mamba block rather than just increased computational power.

3. The paper does not provide solid proof of Mamba’s effectiveness in terms of performance or speed. It omits a comparison with the once leading FCAF3D detector, which scored 71.5 on ScanNet, without explanation. Comparisons of inference speeds across architectures are also absent.

**Questions:**

Besides weakness part, there are some other questions:
1. Some terms could potentially cause confusion. For instance, FPS is an acronym for farthest point sampling, but in the context of the dual Mamba, it refers to farthest point order, while NPS refers to nearest point order.
2. Are there any existing studies that use Mamba in a query-based method? It would be beneficial to reference these works to better understand the uniqueness of this module.

**Limitations:**

Yes.

---

> ### Author Rebuttal · Authors · 2024-08-07
>
> Thanks for your valuable comments.
>
> **Q1**: Comparison of time and space complexity between 3DET-Mamba and other methods and the effects of changing resolution.
>
> **A1**: To demonstrate the superior performance of 3DET-Mamba, we compare the FLOPs and latency of our model with the pervious transformer-based architecture, 3DETR [A], as presented in the Table r1 below. It can be seen that 3DET-Mamba achieves **higher accuracy** (as detailed in Table 1 of our manuscript), **reduced FLOPs** and **lower latency**, as shown by Table r1. In Table 1 of our manuscript, we show that 3DET-Mamba effectively models high-resolution point clouds (i.e., 4096 point clouds), achieving a significant +5.7 mAP@0.5 increase. Furthermore, to explore the impact of changing point resolution, we conduct experiments on changing the density of point clouds. As shown in Figure 1 in the attached pdf file, by **increasing the density of point clouds**, the performance can be continuously **improved** which shows the effectiveness of 3DET-Mamba again.
>
> **Table r1: Performance Comparison of Computational Efficiency**
>
> | Method| FLOPs (↓) | Latency (↓) |
> |-|-|-|
> | 3DETR|14.3|0.22|
> | 3DET-Mamba (our)|9.8|0.13|
>
>
> **Q2**: Additional ablation studies.
>
> **A2**: Thanks for the valuable comment.
> - We first want to clarify that the intention of our work is to explore the feasibility of Mamba in 3D object detection which has not been studied. Previous works mainly explored the usage of Mamba in classification and segmentation tasks. However, this task is challenging due to 1) the non-casual and irregular 3D point clouds hinder the modeling of causal sequences. 2) The original Mamba block is good at extracting global information from long sequences but ignores detailed information to a certain extent which is important in 3D detection tasks.
> - We address the first challenge using ranked queries. To further verify the effectiveness of this approach, we conduct ablation studies shown in Table r2 below, which indicate that discarding ranked queries leads to a decrease in performance.
>
>   **Table r2: Ablation Study on Ranked Queries**
>   | Rank Query | mAP25 | mAP50 |
>   |-|-|-|
>   | ✗ | 65.6|48.1|
>   |✓|66.9|48.7|
>
> - To clarify the relevance of NPS  and FPS in modeling patch sequences, our results in Table r3 demonstrate that combining FPS and NPS can further improve performance by enabling Mamba to models the point cloud in terms of spatial distribution and continuity.
>
>   **Table r3: Impact of NPS and FPS**
>   | NPS | FPS | mAP25 | mAP50 |
>   |-|-|-|-|
>   |✓|✗|65.4|47.7|
>   |✗|✓|65.0|46.3|
>   |✓| ✓|66.9|48.7|
>
> - Existing Mamba-based works mainly use Mamba as the encoder. However, there is still no exploration of using Mamba as a decoder. To explore the potential of Mamba as a 3D decoder, we propose query-based Mamba which can better model the relationship between each query by taking advantage of Mamba. As shown in Table 5 in our article, our query-aware Mamba block can outperform the transformer-based decoder.
>
> - As shown in Table r1, compared to the transformer-based models, our method (which includes the query-aware Mamba block) exhibits lower FLOPs and latency. This demonstrates that the improvements are indeed due to the design of the novel Mamba block, rather than increased computational power.
>
> **Q3**: Comparisons of FCAF3D and speed.
>
> **A3**: Thank you for your valuable comments. In this paper, we focus on indoor 3D object detection using point clouds, a crucial 3D data representation form. Compared to voxels (used in FACF3D), point clouds offer benefits such as high-efficiency storage. We benchmark our 3DET-Mamba against other open-source point-based methods, with results in Table 1 of our manuscript demonstrating our approach's effectiveness. However, it is not trivial to directly apply Mamba to point cloud scenes due to their unordered and non-causal nature. 3DET-Mamba addresses these challenges by introducing a novel local-to-global scanning mechanism and developing the Inner Mamba and Dual Mamba to capture fine-grained point cloud features. Additionally, we designed a Query-aware Mamba Block to decode point cloud information. We hope our work will inspire further use of Mamba as a foundational component for point cloud understanding.
>
> **Q4**: Clarification of FPS and NPS.
>
> **A4**: The FPS and NPS in our article represent the furthest point sampling and the nearest point sampling. The furthest and nearest point order can be obtained after sampling. We will modify the words in dual Mamba to avoid confusion.
>
> **Q5**: Existing methods of query-based mamba works.
>
> **A5**: Thanks for your valuable comments. We carefully investigate query-based mamba works. Currently, there is still no work to explore query-based mamba for object detection. However, some concurrent works introduce query-based mamba for other tasks [B][C]. TM-Mamba[B] modifies the Mamba parameterizes $A$, $B$, $C$, $∆$ as the function of the input and text query to ground the human motion. However, it is not applicable when the number of queries is greater than 1. Besides, QueryMamba[C] combines a query-based transformer decoder and the Mamba encoder. However, it still does not explore query-based Mamba decoder. We will add these related works in the revised version.
>
> [A] Misra I, Girdhar R, Joulin A. An end-to-end transformer model for 3d object detection[C]//Proceedings of the IEEE/CVF international conference on computer vision. 2021.
> [B] Wang X, Kang Z, Mu Y. Text-controlled Motion Mamba: Text-Instructed Temporal Grounding of Human Motion[J].
> [C] Zhong Z, Martin M, Diederichs F, et al. QueryMamba: A Mamba-Based Encoder-Decoder Architecture with a Statistical Verb-Noun Interaction Module for Video Action Forecasting@ Ego4D Long-Term Action Anticipation Challenge 2024[J].

---

> > ### Comment · Reviewer_m4H4 · 2024-08-12
> >
> > Thank you for the author’s rebuttal, which addresses some of my concerns.
> >
> > Is the table r1 fairly comparing 3DETR and 3DET-Mamba with only transformer blocks? I am particularly interested in whether there is an ablation study that demonstrates the necessity of using query-based Mamba rather than cross-attention blocks in a relatively fair comparison, considering the trade-off between computation and accuracy.
> >
> > Given the state of the original paper’s presentation and the missing reference, I can now raise the rating to 5.

---

> > > ### Author Response · Authors · 2024-08-14
> > >
> > > Dear reviewer,
> > >
> > > Thank you for taking the time to review our work and for your valuable comments!
> > >
> > > The results in Table r1 fairly compare 3DETR and 3DET-Mamba in terms of Flops and latency. We further conduct experiments to show the effectiveness and efficiency of Query-aware Mamba (i.e., decoder). Specifically, we replace the Query-aware Mamba with Cross-attention Blocks of the same number of layers. The results are shown in the following table. Importantly, our Query-aware Mamba Blocks not only reduce the computational cost but also further improve detection accuracy.
> > >
> > >
> > >
> > > | **Decoder**                         | **FLOPs (↓)** | **AP@25 (↑)** | **AP@50 (↑)** |
> > > |-------------------------------------|---------------|-----------|-----------|
> > > | Cross-attention Transformer Blocks  | 1.89          | 64.3      | 42.6      |
> > > | Query-aware Mamba Blocks            | **1.76**          | **66.9**      | **48.7**      |
> > >
> > > Note: the FLOPs provided here are specific to the decoder module.
> > >
> > > We will add related works and additional experiments mentioned by the reviewer in the revision of our article. Thanks again for your approval of our work.
> > >
> > > Sincerely,
> > > Authors of paper112

---

### Author Rebuttal · Authors · 2024-08-07

Dear AC and reviewers,

We thank all reviewers (Reviewer m4H4-R1, Reviewer 2ZE6-R2, Reviewer eAYh-R3) for approving our contributions, including **our exploration of mamba for end-to-end indoor 3D object detection for the first time** (R3). The experimental results are **convincing** (R3), demonstrating that the method is **effective** (R1, R2, R3). We also appreciate the acknowledgment of our **clear writing** (R2, R3).


We also thank all reviewers for their insightful feedback to help us improve our paper. We will address all reviewers' concerns and carefully revise the manuscript. **Additional results are provided in the attached PDF file**. We are happy to have further discussion on anything unclear about our paper.

Best regards,
Authors of Paper112

---

### Decision · Program_Chairs · 2024-09-25

**Decision:**

Accept (poster)

**Comment:**

All three reviewers tendentially like the paper and mildly recommend acceptance. The rebuttal helped resolve some of the open issues, and caused one reviewer to raise their rating. Given the technically relevant contribution, the AC recommends acceptance. The authors are encouraged to update their paper in the camera-ready version to include all improvements from the rebuttal.